# Leafy Stems of *Phagnalon saxatile* subsp. *saxatile* from Algeriaas a Source of Chlorogenic Acids and Flavonoids with Antioxidant Activity: Characterization and Quantification Using UPLC-DAD-ESI-MS^n^

**DOI:** 10.3390/metabo11050280

**Published:** 2021-04-29

**Authors:** Farah Haddouchi, Tarik Mohammed Chaouche, Riadh Ksouri, Romain Larbat

**Affiliations:** 1Natural Products Laboratory, Department of Biology, Abou Bekr Belkaid University, B.P 119, Tlemcen 13000, Algeria; tarikmohammed.chaouche@univ-tlemcen.dz; 2Laboratory of Aromatic and Medicinal Plants, Biotechnology Center of Borj-Cédria, B.P 901, Hammam-Lif 2050, Tunisia; riadh.ksouri@cbbc.rnrt.tn; 3Laboratoire Agronomie et Environnement, Université de Lorraine, INRAE, F-54000 Nancy, France; romain.larbat@univ-lorraine.fr

**Keywords:** *Phagnalon saxatile* subsp. *saxatile*, phenolic compounds, polarity, antioxidant activities, UPLC-DAD-ESI-MS^n^, chlorogenic acids

## Abstract

*Phagnalon saxatile* subsp. *saxatile* is a wild species widespread in Algeria which is utilized for medicinal purposes as analgesic and anticholesterolemic. However, information is still scarce regarding its phytochemical content. The objective of this study was to identify and quantify the phenolic compounds from different extracts of its leafy stems. For this purpose, the effects of four extracting solvents were investigated on the content of phenolic compounds and the antioxidant activity of this plant. The extracts prepared with polar solvents (methanol and water) contained higher amounts of phenolic compounds and showed better antioxidant activity than the extracts with apolar solvents (hexane, dichloromethane). The methanolic extract, richest in total phenolic and total flavonoid, had significant antioxidant activity as regarded by DPPH° scavenging capacity (IC_50_ of 5.5 µg/mL), ABTS^+^° scavenging capacity (IC_50_ of 63.8 µg/mL) and inhibition of oxidation of linoleic acid (IC_50_ of 22.7 µg/mL), when compared to synthetic antioxidants. Chlorogenic acids and several flavonoids were identified and quantified by UPLC-DAD-MS^n^. The di-*O*-caffeoylquinic acids isomers were the most concentrated phenolics (25.4 mg/g DW) in the methanolic extract.

## 1. Introduction

The genus *Phagnalon* (Asteraceae) is represented by about 36 species distributed worldwide, six of which are typical of the Mediterranean region [1]. Ethnobotanical studies have reported the use of some species of this genus as food ingredient [2] and popular medicine for the treatment of burns [3], renal lithiasis [4], asthma, headache and toothache [5]. However, studies on this genus are scarce and mainly restricted to only five species: *P. purpurescens* [6], *P. rupestre* [7], *P. sordidum* [4,7], *P. graecum* [8] and *P. saxatile* [9,10,11].

*Phagnalon saxatile* subsp. *saxatile* (*Phagnalon saxatile* (L.) Cass., immortal rocks) is a very common plant, endemic to the Mediterranean region and found in sunny and dry rocky places [12]. Its aerial parts and flowers are traditionally used as an analgesic and to lower blood cholesterol [13]. Moreover, this species has been proposed as a potentially helpful plant to protect against Alzheimer’s disease (AD). This pathology is characterized by oxidative and inflammatory processes leading to neuronal death [14]. The methanolic extract of the flowering aerial parts of *P. saxatile* were shown to exhibit anticholinesterase activity, cholinesterases constituting a master target for treating AD [9]. Interestingly, this property of the *P. saxatile* methanol extract was associated with the anticholinesterase activity of some of its phenolic components, notably caffeic acid, dicaffeoylquinic acid, apigenin and luteolin [6]. The neuroprotective property of *P. saxatile* extract could also be explained by its anti-inflammatory, antioxidant and hypocholesterolemic activities [11]. It was shown that the antioxidant therapy is essential to reduce ROS, which has an impact on neuronal degeneration and that cholesterol is strongly linked to the onset of this disease since high cholesterol increases the risk of AD by three times in old age [14]. From a phytochemical aspect, the aerial part of *P. saxatile* was shown to be a source of an essential oil rich in sesquiterpenoids [15]. In addition, it contains phenolic compounds, namely, 3,3-dimethylallyl-p-benzoquinone [16], 2-alkylhydroquinone glucoside [10], caffeic acid, methylchlorogenic acid, chlorogenic acid, 3,5-dicaffeoylquinic acid and flavonoids such as luteolin, apigenin and apigenin-7-glucoside [9]. Different biological properties were reported for these molecules such as radical-scavenging, antioxidant [17], anti-inflammatory and anti-carcinogenic activities [18].

Phenolic compounds, one of the main classes of secondary metabolites, can be ranged from simple molecules, such as phenolic acids, to highly polymerized compounds, such as tannins. They are phytochemicals including at least one aromatic ring and one or more hydroxyl groups [19]. They are antioxidants and are notably studied in order to find molecules that can protect the human body against the effects of reactive oxygen species (ROS) and replace synthetic antioxidants, such as butylated hydroxytoluene (BHT), butylated hydroxyanisole (BHA) and *tert*-butylhydroquinone (TBHQ) used for many years in food, pharmaceutical and cosmetics industries and which may have a certain toxicity for consumers [20]. These phenolic antioxidants, mainly phenolic acids and flavonoids, have free radicals scavenging, metals chelating and reducing properties while they can also inhibit enzymatic systems responsible for the generation of ROS [21].

This study aimed to investigate the phenolic profile of *P. saxatile* leafy stems (LSPSs), which exhibit significant antioxidant activities (total antioxidant capacity, DPPH· and ABTS^·+^ scavenging ability, reducing power, chelating capacity and β-carotene bleaching test), through LC-MS^n^ analyses. Using extraction solvents with contrasting polarity was a very important step to reach this objective.

## 2. Results

### 2.1. Extract Yields, Polyphenol Contents and Antioxidant Activities

The extract yields were affected by the solvent used, apolar solvents leading to lower values than methanol and water. The apolar solvents also led to poor total phenolic and total flavonoid contents, but revealed the presence of tannins. The contents of total phenolics and total flavonoids were higher for both extracts prepared with polar solvents. The methanolic extract had the highest phenolic (23.1 ± 0.1 mg GAE/g DW) and flavonoid (15.9 ± 0.1 mg CE/g DW) contents (Table 1).

The results of the antioxidant assays for all extracts are summarized in Table 2. Because of their low activity, the hexane and dichloromethane extracts were assayed only for the total antioxidant capacity, DPPH radical scavenging and reducing power, and the values of IC_50_ and EC_50_ were not achieved. Polar solvents (methanol and water), showed important antioxidant capacity compared with standards. The total antioxidant capacity of the extracts, expressed in mg GAE/g DW, increased in proportion to the total phenol content. The DPPH° and ABTS^+^° scavenging capacities of the polar extracts were better than the BHT reference, with the lowest IC_50_ values for the methanolic extract (5.5 ± 0.1 μg/mL and 63.8 ± 0.9 μg/mL, respectively). The methanol extract performed also better than the BHA reference for the β-carotene bleaching assay. However, the ability of the extracts to reduce the iron remained low compared to the BHT (IC_50_ = 99.6 ± 1.8 µg/mL) and ferrous ion chelating activity was much lower than the positive control EDTA, with IC_50_ values reached only for the aqueous extract (4450 ± 3 µg/mL).

### 2.2. Identification and Quantification of Phenolic Compounds Contained in the Polar Extracts of P. saxatile

The methanolic and aqueous extracts that showed the highest antioxidant activity were analyzed by UPLC/MS. Compounds found are given in Table 3, and were numbered by their order of elution. They were characterised based on their UPLC retention time, UV spectra and mostly on their MS^n^ fragmentation. All of these identified compounds were quantified from the UPLC-DAD chromatographic peak areas. The concentrations of hydroxycinnamic acids and flavonoids are expressed in µg/g of the dried weight (DW).
Phenolic compounds identified in methanolic and aqueous extracts of *P. saxatile*.

The UPLC chromatograms at 320 nm of the methanolic and the aqueous extracts of LSPSs are presented in Figure 1A,B, respectively. The phenolic compounds detected were hydroxycinnamic acid esters and flavonoids (Table 3).

#### 2.2.1. Esters of Hydroxycinnamic Acid (Chlorogenic Acids)

**Compounds 1**, **2**, **3** and **5**, detected at 3.7, 5.5, 6.0 and 6.9 min, corresponded to caffeoylquinic acid isomers and the **compounds 8** (tr = 11.0 min), **9** (tr = 11.2 min) and **12** (tr = 12.7 min) were characterized as di-*O*-caffeoylquinic acid isomers [22,23]. **Compound 6**, detected only in methanolic extract, was eluted at 7.2 min and had a *m*/*z* at 337.0924 [M − H]^−^. Its exact mass, calculating at 338.10034, corresponded to coumaroylquinic acid [22,23].

#### 2.2.2. Flavonoids

In addition to the phenolic acid derivatives described above, some flavonoids were also identified in methanolic and aqueous LSPSs extracts. In the aqueous extract, all flavonoids were found in the form of derivatives generally glycosylated. Free aglycones were found in methanolic extract, in addition to the glycosylated forms.

Flavonol derivatives: **Compound 7**, detected at 8.9 min, corresponded to luteolin di-glucoside [24], which had a *m*/*z* at 609.1428 [M − H]^−^. It had an exact mass equal to 610.1507. The MS^2^ fragmentation showed the aglycone ion Y^−^_0_ at 285 [luteolin − H]^−^ and also a fragment ion at *m*/*z* 447. **Compound 10** (11.3 min) in the aqueous extract, **13** (13.3 min) in the two extracts correspond to luteolin glucoside [25] and had a [M − H]^−^ ion at *m*/*z* 447.0904. They presented an exact mass equal to 448.0983 and had characteristic fragment ion at *m*/*z* 285 [luteolin − H]^−^. **Compound 11**, eluted at 11.4 min, presented an exact mass of 464.09375 and displayed a [M-H]^−^ ion at *m*/*z* 463.0855. It corresponded to quercetin glucoside [25]. The MS^2^ spectrum showed the aglycone ion Y^−^_0_ at *m*/*z* 300 [quercetin − H]^−^ corresponding to the loss of hexose residue (162 Da).

Flavonol aglycone: In the methanolic extract, **compound 14** detected at 15.8 min, corresponds to the luteolin and had a mass at *m*/*z* 285.0385 [M − H]^−^ [26]. Its exact mass was 286.0464. The fragment at *m*/*z* 151(^1,3^A^−^) and *m*/*z* 133 (^1,3^B^−^) in negative mode correspond to the type of fragmentation Retro Diels-Alder (RDA) of aglycone flavonols [27].

Flavone aglycone: **Compound 15** (16.5 min) was detected in the methanolic extract. It corresponds to the apigenin having a *m*/*z* at 269.0444 [M − H]^−^. It had an exact mass equal to 270.0523 [26]. The fragment at *m*/*z* 151 (^1,3^A^−^) in the negative mode corresponded to a fragmentation Type Retro Diels-Alder (RDA).

#### 2.2.3. Unknown Compound

**Compound 4** was unknown and was eluted at 6.1 min in both extracts.
UPLC quantification of phenolic acid derivatives and flavonoids

Phenolics from the methanolic and aqueous extracts were quantified (Table 3). The total amounts of compounds (mg/g DW) were calculated by summarizing individual amounts of all constituents (Table 4). Di-caffeoylquinic acid isomers were the most concentrated in the methanolic extract (25.37 mg/g DW), followed by caffeoylquinic acid isomers (12.56 mg/g DW), flavonoids (4.38 mg/g DW) and coumaroylquinic acid (0.07 mg/g DW) (Table 4). The phenolic concentrations in the aqueous extract were 3 to 10-fold lower than in the methanolic extract for caffeoylquinic acids and dicaffeoylquinic acids, respectively. Interestingly, the aqueous extract was more concentrated in caffeoylquinic acids (3.90 mg/g DW) than the di-caffeoylquinic acids (2.80 mg/g DW).

Regarding flavonoids, the luteolin glucoside and quercetin glucoside were the most abundant in the extracts. The methanolic extract contained a higher diversity of flavonoids with a globally higher concentration, whereas only luteolin di-glucoside was more concentrated in the aqueous extract. Free aglycones were only found in the methanolic extract but at low concentration (luteolin at 0.13 mg/g DW and apigenin at 0.03 mg/g DW; Table 4).

The distribution between phenolic acids and flavonoids were 90% and 10%, respectively, in the methanolic extract and 76% and 24% in the aqueous extract (Table 4).

## 3. Discussion

The present work presented the phenolic characterization and antioxidant properties of different LSPSs extracts. The nature and polarity of the solvent used for the extraction impacted directly the yield and extractability of phenolic compounds. This is in agreement with previous studies that have demonstrated that the yields of phenolic compounds obtained with polar solvents were higher than those of less polarity [8,9,28,29]. Unexpectedly, the total tannin assay led to detect the presence of a small fraction of tannins in the apolar fractions only. Since hexane and dichloromethane extraction were previously shown to extract tannins with low efficiency [30,31] the fact that no trace of tannin was detected in the polar fraction is puzzling. UPLC analyses confirmed the absence of tannins in the polar extractions, however, since UPLC analyses were not conducted on the apolar fractions, we cannot strictly confirm the nature of the compound(s) that led to a positive result with the total tannin assay.

To our knowledge, a few studies have investigated the phenolic composition and antioxidant properties on *Phagnalon* species, including those carried out on *P. saxatile* from Italy [9] and Algeria [10,11], *P. graecum* Boiss. from Turkey [8] and *P. lowei* from Madeira archipelago [32]. The antioxidant activities measured in our methanolic extract were close to the Italian *P. saxatile* extract, regarding the DPPH and β-carotene bleaching test [9]. Accordingly to the study on *P. graecum*, the LSPSs aqueous extract appeared more active than the other solvent for the iron chelating activity [8].

In total, eight phenolic compounds were characterized and quantified in the LSPSs methanolic and aqueous extracts, including flavonoids and three hydroxycinnamic acid esters using UPLC-DAD-ESI-MS^n^ analysis. Hydroxycinnamic acid esters, more precisely caffeoyl and dicaffeoylquinic acids were predominantly detected. These metabolites have already been detected in other *Phagnalon* species [9,10,33,34] and appeared as the main phytochemical components in a diversity of Asteraceae [32,35]. Quantitatively, caffeoyl and dicaffeoylquinic acids were the most abundant phenolic compounds in LSPSs extracts analyzed. Their total concentration in the methanolic extract reached 38 mg/g DW (i.e., 3.8 g/100 g) which ranks them in the upper part of the caffeoyl-quinic derivative-accumulating plants, far below *Ilex paraguariensis* (herbal mate, 9.2 g/100 g) but in the same range as Arabica coffee beans (4.9 g/100 g) and above white and green tea (*Camilla sinensis*, 1.3–1.6 g/100 g) [36,37,38,39]. In addition, the concentrations in caffeoyl and dicaffeoyl-quinic acids measured in our *P. saxatile* samples are similar to the concentrations measured in the methanol extracts of *P. lowei* which was highlighted as the richest for these compounds on a panel of 10 Asteraceae [32]. Interestingly, in the same study, the authors identified that the *P. lowei* extract had the best inhibitory activities on a set of digestive enzymes (α-amylase, α- and β-glucosidases, lipase) that are related to diabetes and obesity control, which appeared to be linked to the concentration of caffeoylquinic acid derivatives. Considering this study, a future study should be conducted on the beneficial impact of *P. saxatile* in the control of diabetes and obesity.

Flavonols and flavones were also identified in this plant, showing interesting, concentrations for luteolin glucosides and quercetin glucosides. The presence of aglycone flavonoids, only in the methanolic extract, can be due to their solubility, considered to be strongly affected by the nature of both the solvent and the flavonoid structure. There is, however, very limited data on direct measurement of solubility of differently substituted flavonoids, which are obtained by comparing extraction or chromatography data, such as, for example, comparisons of glycosylated and non-glycosylated compounds. The presence of a sugar moiety usually increases the solubility of the flavonoids in polar solutions. For example, glycosylated flavonols such as glycosylated quercetin have considerably better water solubility compared to the corresponding aglycone [40]. The position of the sugar moiety and the number of the bound sugars also play important roles. For example, the diglucoside quercetin is more stable than the monoglucoside [41].

From the results of quantification, we can notice that the amount of total phenolic compounds obtained by the UPLC analysis was different from that estimated by the Folin-Ciocalteu method, which can be explained by the limitation of the phenolic substances used as standard [42]. However, the absence of tannins by the assay was confirmed by UPLC.

The relationship between the antioxidant activity and phenolic composition of the extracts is a valid approach to understand the activity of the extracts of *P. saxatile*. It was previously reported that cinnamic acid derivatives display a high antioxidant capacity and that the caffeoyl group is essential to the radical scavenging activity and lipid anti-peroxidation [7,43]. Several studies have shown that the isomers of caffeoylquinic and di-caffeoylquinic acids, the major components identified in LSPSs extracts, have antioxidant activity with a higher activity for the latter one [43,44,45]. This property is in accordance with the high antioxidant activity measured in our methanolic extracts. These natural antioxidants have commercial applications in medicine, food, and cosmetics [46]. Their medical interest lies in their various biological activities such as anti-oxidative ability, antiviral, antibacterial, anti-inflammatory, antispasmodic activities, reduction in the relative risk of cardiovascular disease and type 2 diabetes, inhibition of the mutagenicity of carcinogenic compounds [47].

Additionally, it was shown that these molecules, but not caffeine, confer a significant protection against oxidative stress, in vitro, and could be responsible for the beneficial health effects associated with the consumption of green coffee bean derived products, such as extracts commercialized as nutraceuticals or soluble green coffee either alone or blended with roasted coffee [48]. Therefore, regarding their high concentration in these molecules, the extracts of LSPSs could be used as a primary source in the supply chain of the cosmetic, pharmaceutical or health industries which uses chlorogenic acids as active principles. In addition, LSPS extracts could also be used to generate an innovative functional product, such as food supplements, as is the case for coffee-derived food supplements, which have attracted a lot of attention for their promise of quick weight loss [49].

Aglycones and glycosylated flavonoids present in the analysed extracts are of great general interest due to their antioxidant and chelating abilities. Their structure-activity relationships as antioxidants were comprehensively studied. It was previously shown that aglycones are more potent antioxidants than their corresponding glycosides and that flavonoids are able to reduce highly reactive free radicals through the conjugations and hydroxyl groups in particular in position 3′ of the B ring and 3 of the C ring. They are able to inhibit a wide range of enzymes generating ROS, have a reducing power and can also chelate catalysts of oxidation reactions such as metal ions (iron and copper). Several studies have shown the antioxidant capacity of quercetin and luteolin [45,50,51,52,53].

## 4. Materials and Methods

### 4.1. Plant Material and Extract Preparation

The flowering aerial parts of *P. saxatile* were collected in March 2011 from Tlemcen, in the west of Algeria. They were identified in the Laboratory of Natural Products, Department of Biology, University of Tlemcen, Algeria. Voucher specimens were deposited at the Herbarium of the Laboratory.

The plant parts were dried at room temperature for two weeks. Ten grams of fine powder of the leafy stems were extracted successively with hexane, dichloromethane, methanol and water, in a Soxhlet apparatus, following the protocol as already described by Chaouche et al. [29].

### 4.2. Determination of Total Polyphenol, Total Flavonoid, and Total Tannin Contents

To select the extracts rich in phenolic compounds, the contents of total polyphenols, flavonoids and tannins in the hexane, dichloromethane, methanol and water extracts, were determined. Estimation of different phenolic class contents was realized as already described in our previous paper [11]. The total phenolic content was determined using the Folin-Ciocalteu reagent and was expressed with reference to the gallic acid standard curve (Y = 0.00253 x + 0.01344, R^2^ = 0.995). Total flavonoid content was determined using sodium nitrite and aluminum chloride and was expressed with reference to the catechin standard curve (Y = 0.00355 x + 0.09903, R^2^ = 0.987). The total tannin content was determined using a vanillin-methanol solution and was expressed with reference to the catechin standard curve (Y = 0.00076 x − 0.0076, R^2^ = 0.994).

### 4.3. Determination of Antioxidant Activities

To evaluate the antioxidant activity, in vitro, six chemical tests were realized following the procedures described in Haddouchi et al. [11]: Total antioxidant capacity, 1,1-diphenyl-2-picrylhdrazyl (DPPH); 2,2′-azino-bis (3-ethylbenzthiazoline-6-sulphonic acid) (ABTS); Iron reducing power; β-carotene bleaching and chelating effect on ferrous ions.

For scavenging DPPH and ABTS radicals, chelating power and β-Carotene bleaching methods, lower IC_50_ value (the concentration required to cause a 50% inhibition) indicates a higher antioxidant activity. For reducing power, lower EC_50_ value (the effective concentration giving an absorbance of 0.5) indicates a higher antioxidant activity.

### 4.4. LC-DAD-ESI-MS^n^ Analysis

The phenolic composition of the extracts (50 µL) was analyzed on an HPLC-DAD-ESI-MS^n^ system composed by a binary solvent delivery pump (Ultimate 3000, Thermo Scientific-Dionex, Bremen, Germany) connected to a diode array detector (Surveyor PDA plus, Thermo-Finnigan, USA) and an LTQ Orbitrap mass spectrometer (Thermo Scientific, Bremen, Germany) according to the method reported by Chougui et al. [54]. Briefly, the molecules were separated on a C18 LichroCART (250 mm × 4.6 mm) column (Merck, Darmstadt, Germany) using water formic acid 0.1% as solvent A and MeOH, acetonitrile, formic acid (50/50/0.1) as solvent B. The separation method consisted of an isocratic step at 1% of B for 2 min followed by a linear gradient from 1% to 40% of B in 100 min, then to 70% of B in 25 min and 90% of B in 55 min with a flow rate at 700 µL/min. The mass spectrometer conditions and data processing were the same as described by Chougui et al. [55]. Experimental exact masses and MS^2^ fragmentation data were compared to metabolomics data banks (Pubchem Compound: https://pubchem.ncbi.nlm.nih.gov/, accessed on 21 February 2021; ReSpect: https://spectra.psc.riken.jp/, accessed on 21 February 2021; Mass Bank: https://www.massbank.jp, accessed on 21 February 2021) and available literature to identify the nature of the phenolic compounds.

The phenolic quantification was realized on a U-HPLC system (Shimadzu, Kyoto, Japan) consisting of a binary solvent delivery pump connected to a diode array detector. Three to five microliters of the extracts were separated on a C18 Kinetex (100 mm × 2.1 mm) column (Phenomenex, Torrance, CA, USA) by using a gradient elution from 10 to 45% MetOH for 13 min, then 99% MetOH for 3 min with a flow rate of 300 µL min^−1^. The column was rinsed for 1 min with 99% MetOH and re-equilibrated to 10% MetOH for 2 min prior to the next run. Compound quantification was based on measurement of area under each peak determined at 320 nm and 350 nm and expressed relative to calibration curves with chlorogenic acid (for caffeoyl and dicaffeoylquinic isomers), coumaric acid (for coumaroyl quinic acid), apigenin (for aglycon and mono-glycosylated flavonoids) and rutin (for diglycosylated flavonoids).

### 4.5. Statistical Analysis

All measurements were performed in triplicates and results are expressed as mean ± standard deviation (SD) and were subjected to analysis of variance (ANOVA). Differences were considered to be significant at the level of *p* < 0.05.

## 5. Conclusions

*P. saxatile* showed important antioxidant activity. The methanolic extract of leafy stems had the highest amounts of phenolic compounds and showed stronger antioxidant capacities regarding the DPPH, ABTS and β-carotene bleaching assays.

Phenolic compounds were identified and quantified in this plant from Algeria, for the first time, by the reliable and reproducible UPLC-DAD-ESI/MS^n^ method. LSPSs constitutes an important and abundant source of caffeoylquinic and dicaffeoylquinic acids isomers which has demonstrated pharmacological properties.

Therefore, these results may provide useful information for researchers in phytopharmacy and phytotherapy because of the protective roles of these molecules in pathologies associated to oxidative stresses.

Additionally, this study revealed the presence of important metabolites that would allow emphasising unknown species such as *P. saxatile* under a nutritional aspect. It will be interesting to explore the potential of LSPSs extracts as a functional supplement in the food industry. This aspect deserves further studies to determine the maximum-tolerated dose and potential toxicity of this plant for human.

## Figures and Tables

**Figure 1 metabolites-11-00280-f001:**
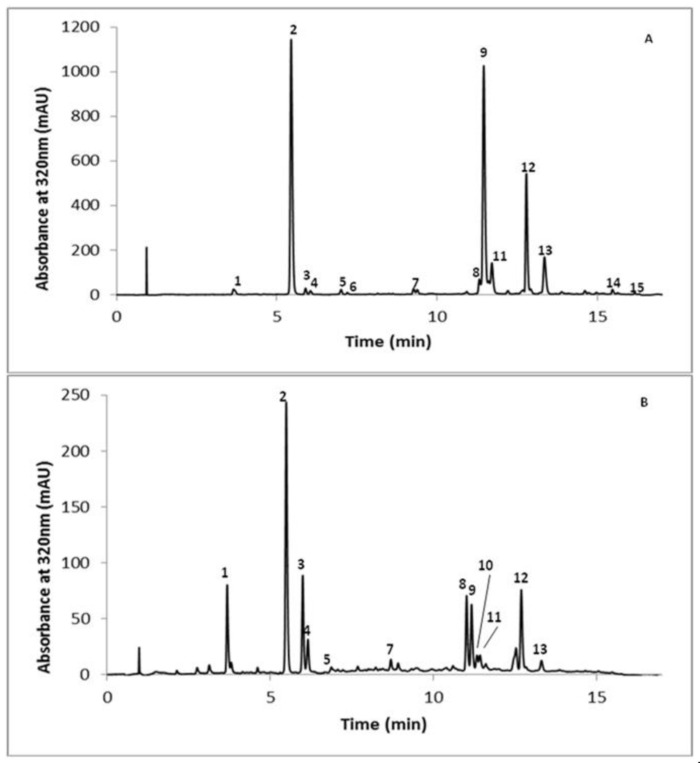
Chromatographic profiles of LSPSs methanolic extract (**A**) and LSPSs aqueous extracts (**B**). The chromatographic profiles are followed at 320 nm which corresponds to a wavelength where both phenolic esters and flavonoids can be detected. The numbers on the peaks are related to their identification described in Table 3.

**Table 1 metabolites-11-00280-t001:** The yields and phenolic contents (TPC, TFC and TTC) of extracts obtained with four extracting solvents of different polarity (hexane, dichloromethane, methanol and water).

	Extract	Yield (%)	TPC (mg GAE/g DW)	TFC (mg CE/g DW)	TTC (mg EC/g DW)
**LSPSs**	**Hx**	**2.3**	0.50 ± 0.03	0	0.60 ± 0.02
	**D**	**0.8**	0.40 ± 0.01	0.30 ± 0.01	0.40 ± 0.02
	**M**	**8.2**	23.1 ± 0.1	15.9 ± 0.1	0
	**W**	**8.5**	15.1 ± 0.3	8.2 ± 0.1	0

LSPSs: Leafy Stem of Phagnalon saxatile. Hx: Hexane, D: Dichloromethane, M: Methanol, W: water, TPC (total phenolic content), TFC (total flavonoid content), TTC (total tannin content).

**Table 2 metabolites-11-00280-t002:** Antioxidant properties of LSPSs extracts. The antioxidant properties are analyzed through 6 measurements including total antioxidant capacity, DPPH· and ABTS^·+^ scavenging ability, reducing power, chelating capacity and β-carotene bleaching test.

	Extract	Total Antioxidant Activity (mg EAG/g MS)	IC_50/DPPH_ (µg/mL)	EC_50/iron reducing_ (µg/mL)	IC_50/ABTS_ (µg/mL)	IC_50/Iron chelation_ (µg/mL)	IC_50/__β__-carotène_ (µg/mL)
**LSPSs**	**Hx**	1.7 ± 0.1	-	-	/	/	/
	**D**	0.4 ± 0.1	-	-	/	/	/
	**M**	18.2 ± 0.3	5.5 ± 0.1	343 ± 5	63.8 ± 0.9	-	22.7 ± 0.4
	**W**	11.8 ± 0.1	7.0 ± 0.1	332 ± 2	122.9 ± 1.8	4450 ± 3	125.9 ± 1.5
**Standards**			10.5 ± 0.4 ^a^	99.6 ± 1.8 ^a^	73.1 ± 1.7 ^a^	46.5 ± 0.3 ^b^	48 ± 0.9 ^c^

LSPSs: Leafy Stem of Phagnalon saxatile,/: not tested, -: IC50 value not achieved, Hx: Hexane, D: Dichloromethane, M: Methanol, W: water, ^a^ BHT, ^b^ EDTA, ^c^ BHA.

**Table 3 metabolites-11-00280-t003:** Identification and quantification of phenolic compounds contained in the methanolic and the aqueous extracts of *P. saxatile*, by HPLC-DAD-ESI/MS^n^.

Peak	Rt	λ Max (nm)	MS^2^ [M − H]^−^	[M − H]^−^	Calculated Mass	Formula	Compound Name	LSPSs M (μg/g DW)	LSPSs Aq (μg/g DW)
1	3.7	300, 324	353, 191, 179, 135	353.0864	354.0943	C_16_H_18_O_9_	Caffeoylquinic acid isomer	326.0 ± 20.8	694.0 ± 7.0
2	5.5	239, 300, 324	353, 191, 179, 135	353.0864	354.0943	C_16_H_18_O_9_	Caffeoylquinic acid isomer	11,807.0 ± 1000.6	2373.3 ± 24.8
3	6.0	239, 300, 326	353, 191, 179, 135	353.0864	354.0943	C_16_H_18_O_9_	Caffeoylquinic acid isomer	230.3 ± 26.6	793.7 ± 19.0
4	6.1						NI		
5	6.9	239, 300, 326	353, 191, 179, 135	353.0864	354.0943	C_16_H_18_O_9_	Caffeoylquinic acid isomer	199.0 ± 16.5	56.3 ± 2.9
6	7.2	300 sh, 323	353, 191, 179	337.0926	338.1003		Coumaroyl quinic acid	73.7 ± 5.5	-
7	8.9	330	609, 447, 285	609.1437	610.1507		Luteolin di-glucoside	290.0 ± 19.9	529.7 ± 29.1
8	11.0	297, 326	515, 353, 335, 179, 173	515.1106	516.1185	C_25_H_24_O_12_	di-*O*-Caffeoylquinic acid isomer	921.3 ± 69.3	945.0 ± 9.5
9	11.2	297, 326	515, 353, 335, 179, 173	515.1106	516.1185	C_25_H_24_O_12_	di-*O*-Caffeoylquinic acid isomer	17,214.7 ± 1824.3	810.7 ± 15.0
10	11.3	346	447, 285	447.0904	448.0983	C_21_H_20_O_11_	Luteolin glucoside	-	683.5 ± 6.9
11	11.4	251, 350	463,300	463.0855	464.0934	C_21_H_20_O_12_	quercetin-glucoside	2220.6 ± 178.9	554.0 ± 16.8
12	12.7	300, 328	515, 353, 335, 179, 173	515.1166	516.1245	C_25_H_24_O_12_	di-*O*-Caffeoylquinic acid isomer	7230.0 ± 603.9	1077.7 ± 14.6
13	13.3	267, 336	447, 285	447.0904	448.0983	C_21_H_20_O_11_	Luteolin glucoside	1709.9 ± 134.9	378.8 ± 33.9
14	15.8	254, 350	285, 133, 151	285.0385	286.0464	C_15_H_10_O_6_	Luteolin	130.3 ± 12.4	-
15	16.5	267, 342	269, 151	269.0444	270.0523	C_15_H_10_O_5_	Apigenin	30.0 ± 2.7	-

The peak numbers follow their retention time (Rt) and are the same as the ones in Figure 1. The proposed formulas and names are based on the maximum absorbance spectrum (λ max), the *m*/*z* ratio of the parent and fragment ions detected on negative mode ([M − H]^−^). NI: not identified, -: not detected.

**Table 4 metabolites-11-00280-t004:** Amounts and percentages of phenolic compounds contained in methanolic and aqueous extracts of *P. saxatile*, identified by HPLC-DAD-ESI/MS^n^.

Class of Phenolic Compounds	Compound Name	LSPSs M (mg/g DW)	LSPSs Aq (mg/g DW)
hydroxycinnamic acid esters	Caffeoylquinic acid isomer	12.56	3.90
	di-*O*-Caffeoylquinic acid isomer	25.37	2.80
	Coumaroyl quinic acid	0.07	-
	***Sum of individual amounts***	***38.00***	***6.70***
	**% of the total phenolic compounds**	**90%**	**76%**
Flavonoids	Luteolin di-glucoside	0.29	0.53
	Luteolin glucoside	1.71	1.06
	Luteolin	0.13	-
	Quercetin-glucoside	2.22	0.55
	Apigenin	0.03	-
	***Sum of individual amounts***	***4.38***	***2.14***
	**% of the total phenolic compounds**	**10%**	**24%**

-: not detected.

## Data Availability

All data are contained within the article.

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
