# Peer review of "Leafy Stems of Phagnalon saxatile subsp. saxatile from Algeriaas a Source of Chlorogenic Acids and Flavonoids with Antioxidant Activity: Characterization and Quantification Using UPLC-DAD-ESI-MSn"

_metabolites, 2021, doi:10.3390/metabo11050280_

Round 1
Reviewer 1 Report
The manuscript entitled "Leafy stems of Phagnalon saxatile subsp. saxatile from Algeria as a source of chlorogenic acids and flavonoids with antioxidant activity: Characterization and quantification using UPLC-DAD-ESI-MSn" by Haddouchi et al. shows a phytochemical work on a few investigated Mediterranean species, accompained by preliminary and basic cell-free biological essays.
The manuscript, in my opinion, has some strenghts, but big limits, as well.
Strenghts: the species is worthy to be better investigated and to be taken into account in scientific research; anti-radicalic results are sounding; the phytochemical characterization is well designed.
Limits:
Major concerns:
1- Authors should add some information about the ethnobotanical use of the species, traditional preparations, eventual safety concerns, and they should better link the few known tradition to the hypothetical and desirable modern use with health purposes. Indeed, the species is not enlisted in pharmacopoeias and in botanical food supplements lists and the putative interest for human health is not immediate if not adequately described.
2-Linked to the point above: authors should discuss data and obtained results by comparing data with those related to similar tests performed on different and known species, in order to better understand novelty, scientific soundness and perspectives.
3- Authors know that rutin is not a perfect standard to quantify total flavonoids when only monoglycosides and aglycones are present. Actual values could be really overrated if rutin is used; I suggest to express flavonoids as kaempferol.
4- recorded lambda max of many phenolics are strange... check carefully (for example kaempferol aglycone has a maximum > 360 nm, kaempferol glucosides at 265-275 nm...)
Minor concerns:
check English language;
use same precision alongside the paper (RT 6.0 for example, not, 6...)
Write the species in the right form the first time:
Phagnalon saxatile subsp. saxatile (syn= Phagnalon saxatile (L.) Cass. )
Author Response
Major concerns:
1- Authors should add some information about the ethnobotanical use of the species, traditional preparations, eventual safety concerns, and they should better link the few known tradition to the hypothetical and desirable modern use with health purposes. Indeed, the species is not enlisted in pharmacopoeias and in botanical food supplements lists and the putative interest for human health is not immediate if not adequately described.
Additional informations have been added in the introduction regarding the usage of plants of the Phagnalon genus in general and P. saxatile in particular.
The elements added/modified are :
Line 33 : Ethnobotanical studies have reported the use of some species of this genus as food ingredient [2] and popular medicine for the treatment of burns [3] , renal lithiasis [4], asthma, headache and toothache [5]. However, studies on this genus are scarce and mainly restricted to…
Line 42 : Moreover, this species as been proposed as a potentially helpful plant to protect against Alzheimer’s disease (AD). This pathology is characterized by oxidative and inflammatory processes leading to neuronal death [14]. The methanolic extract of the flowering aerial parts of P. saxatile were shown to exhibit anticholinesterase activity, cholinesterases constituting a master target for treating AD [9]. Interestingly, this property of the P. saxatile methanol extract was linked to the anticholinesterase activity of some of its phenolic components, notably caffeic acid, dicaffeoyl quinic acid, apigenin and luteolin [6]. The neuroprotective property of P. saxatile extract could also be explained by their anti-inflammatory, antioxidant and hypocholesterolemic activities [11]. It has been shown that the antioxidant therapy is vital in ROS preventing neuronal degeneration and that cholesterol is strongly linked to the onset of this disease since high cholesterol increases the risk of AD three times in old age [14].
2-Linked to the point above: authors should discuss data and obtained results by comparing data with those related to similar tests performed on different and known species, in order to better understand novelty, scientific soundness and perspectives.
We have taken in to account this remark despite the fact that a few studies are available on the Phagnalon genus. Informations have been added in order to compare P. saxatile content in caffeoylderivativeswithotherknown plants accumulatingthese compounds. In addition, wediscuss the impact of the high concentration of these compounds withtheirinterest in the control of diabete and obesity. The modification are shownbelow :
Line 203 : To our knowledge, a few studies have investigated on the phenolic composition and antioxidant properties on Phagnalon species, including those carried out on P. saxatile from Italy [9] and Algeria [10,11], P. graecum Boiss. from Turkey [8] and P. lowei from Madeira archipelago [32]. The antioxidant activities measured in our methanolic extract were close to the Italian P. saxatile extract, regarding the DPPH and β-carotene bleaching test [9]. Accordingly to the study on P. graecum, the LSPSs aqueous extract appeared more active than the other solvent for the iron chelating activity [8].
Line 213 : These metabolites have already been detected in other Phagnalon species [9,10,33,34] and appeared as the main phytochemical components in a diversity of Asteracea [32,35]. Quantitatively, caffeoyl and dicaffeoylquinic acids were the most abundant phenolic compounds in LSPSs extracts analyzed. Their total concentration in the methanolic extract reached 38 mg/g DW (i.e 3.8 g/100g) which ranks them in the upper part of the caffeoyl quinic derivative-accumulating plants, far below Ilex paraguariensis (herbal mate, 9.2 g/100g) but in the same range as Arabica coffee beans and above white and green tea (Camilla sinensis, 13-16 mg/100g) [36-39]. In addition, the concentrations in caffeoyl and dicaffeoylquinic acids measured in our P. saxatile samples are similar to the concentrations measured in the methanol extracts of P. lowei which was highlighted as the richest for these compounds on a panel of 10 Asteraceaes [32]. Interestingly, on the same study, the authors identified that the P. lowei extract had the best inhibitory activities on a set of digestive enzymes (a-amylase, a- and b-glucosidase, lipase) that are related to diabete and obesity control which appeared to be linked to the concentration of caffeoylquinic acid derivatives. Considering this study, future prospect should be done on the beneficial impact of P. saxatile in the control of diabete and obesity.
3-Authors know that rutin is not a perfect standard to quantify total flavonoids when only monoglycosides and aglycones are present. Actual values could be really overrated if rutin is used; I suggest to express flavonoids as kaempferol.
We agree with reviewer regarding the choice of the flavonoid standard. Now, we are expressing the concentrations of aglycone and monoglycosylatedflavonoids relative to an apigenin standard curve. Diglycosylatedflavonoid are stillexpressed relative to a rutin standard curve.
4- recorded lambda max of many phenolics are strange... check carefully (for example kaempferol aglycone has a maximum > 360 nm, kaempferol glucosides at 265-275 nm...)
This part has been extensively checked. We have detected errors regarding the nature of some flavonoids. Indeed, the initial attribution to kaempferol moiety has been revised and requalified as luteolin moiety. This is in agreement with (i) the lambda max <350nm, (ii) the presence of a MS2 fragment at m/z 133 which is specific for luteolin and not kampferol. In addition, these data fit well with previous studies on P.saxatile describing luteolin derivatives but not kaempferol (Conforti et al., 2010 ; Cherchar et al., 2018)
Minor concerns:
check English language; The document has been checked and corrected
usesameprecisionalongside the paper (RT 6.0 for example, not, 6...) This has been checked and corrected.
Write the species in the right form the first time: Phagnalon saxatile subsp. Saxatile (syn= Phagnalon saxatile (L.) Cass.)
This has been modified
Major concerns:
1- Authors should add some information about the ethnobotanical use of the species, traditional preparations, eventual safety concerns, and they should better link the few known tradition to the hypothetical and desirable modern use with health purposes. Indeed, the species is not enlisted in pharmacopoeias and in botanical food supplements lists and the putative interest for human health is not immediate if not adequately described.
Additional informations have been added in the introduction regarding the usage of plants of the Phagnalon genus in general and P. saxatile in particular.
The elements added/modified are :
Line 33 : Ethnobotanical studies have reported the use of some species of this genus as food ingredient [2] and popular medicine for the treatment of burns [3] , renal lithiasis [4], asthma, headache and toothache [5]. However, studies on this genus are scarce and mainly restricted to…
Line 42 : Moreover, this species as been proposed as a potentially helpful plant to protect against Alzheimer’s disease (AD). This pathology is characterized by oxidative and inflammatory processes leading to neuronal death [14]. The methanolic extract of the flowering aerial parts of P. saxatile were shown to exhibit anticholinesterase activity, cholinesterases constituting a master target for treating AD [9]. Interestingly, this property of the P. saxatile methanol extract was linked to the anticholinesterase activity of some of its phenolic components, notably caffeic acid, dicaffeoyl quinic acid, apigenin and luteolin [6]. The neuroprotective property of P. saxatile extract could also be explained by their anti-inflammatory, antioxidant and hypocholesterolemic activities [11]. It has been shown that the antioxidant therapy is vital in ROS preventing neuronal degeneration and that cholesterol is strongly linked to the onset of this disease since high cholesterol increases the risk of AD three times in old age [14].
2-Linked to the point above: authors should discuss data and obtained results by comparing data with those related to similar tests performed on different and known species, in order to better understand novelty, scientific soundness and perspectives.
We have taken in to account this remark despite the fact that a few studies are available on the Phagnalon genus. Informations have been added in order to compare P. saxatile content in caffeoylderivativeswithotherknown plants accumulatingthese compounds. In addition, wediscuss the impact of the high concentration of these compounds withtheirinterest in the control of diabete and obesity. The modification are shownbelow :
Line 203 : To our knowledge, a few studies have investigated on the phenolic composition and antioxidant properties on Phagnalon species, including those carried out on P. saxatile from Italy [9] and Algeria [10,11], P. graecum Boiss. from Turkey [8] and P. lowei from Madeira archipelago [32]. The antioxidant activities measured in our methanolic extract were close to the Italian P. saxatile extract, regarding the DPPH and β-carotene bleaching test [9]. Accordingly to the study on P. graecum, the LSPSs aqueous extract appeared more active than the other solvent for the iron chelating activity [8].
Line 213 : These metabolites have already been detected in other Phagnalon species [9,10,33,34] and appeared as the main phytochemical components in a diversity of Asteracea [32,35]. Quantitatively, caffeoyl and dicaffeoylquinic acids were the most abundant phenolic compounds in LSPSs extracts analyzed. Their total concentration in the methanolic extract reached 38 mg/g DW (i.e 3.8 g/100g) which ranks them in the upper part of the caffeoyl quinic derivative-accumulating plants, far below Ilex paraguariensis (herbal mate, 9.2 g/100g) but in the same range as Arabica coffee beans and above white and green tea (Camilla sinensis, 13-16 mg/100g) [36-39]. In addition, the concentrations in caffeoyl and dicaffeoylquinic acids measured in our P. saxatile samples are similar to the concentrations measured in the methanol extracts of P. lowei which was highlighted as the richest for these compounds on a panel of 10 Asteraceaes [32]. Interestingly, on the same study, the authors identified that the P. lowei extract had the best inhibitory activities on a set of digestive enzymes (a-amylase, a- and b-glucosidase, lipase) that are related to diabete and obesity control which appeared to be linked to the concentration of caffeoylquinic acid derivatives. Considering this study, future prospect should be done on the beneficial impact of P. saxatile in the control of diabete and obesity.
3-Authors know that rutin is not a perfect standard to quantify total flavonoids when only monoglycosides and aglycones are present. Actual values could be really overrated if rutin is used; I suggest to express flavonoids as kaempferol.
We agree with reviewer regarding the choice of the flavonoid standard. Now, we are expressing the concentrations of aglycone and monoglycosylatedflavonoids relative to an apigenin standard curve. Diglycosylatedflavonoid are stillexpressed relative to a rutin standard curve.
4- recorded lambda max of many phenolics are strange... check carefully (for example kaempferol aglycone has a maximum > 360 nm, kaempferol glucosides at 265-275 nm...)
This part has been extensively checked. We have detected errors regarding the nature of some flavonoids. Indeed, the initial attribution to kaempferol moiety has been revised and requalified as luteolin moiety. This is in agreement with (i) the lambda max <350nm, (ii) the presence of a MS2 fragment at m/z 133 which is specific for luteolin and not kampferol. In addition, these data fit well with previous studies on P.saxatile describing luteolin derivatives but not kaempferol (Conforti et al., 2010 ; Cherchar et al., 2018)
Minor concerns:
check English language; The document has been checked and corrected
usesameprecisionalongside the paper (RT 6.0 for example, not, 6...) This has been checked and corrected.
Write the species in the right form the first time: Phagnalon saxatile subsp. Saxatile (syn= Phagnalon saxatile (L.) Cass.)
This has been modified

Reviewer 2 Report
The manuscript metabolites 1137365 is good work and analytically classic. In general terms the manuscript is fine. Table 1 reports a tannin content in the hexane and dichloromethane fractions, which does not seem correct to me. The method used to measure tannins tends to give false positives. But there is also the possibility that the plant may contain methoxylated flavonoids, which do appear in the aforementioned fractions. The authors should check this information against a chromatographic run of these fraction
Author Response
Reviewer 2
The manuscript metabolites 1137365 is good work and analytically classic. In general terms the manuscript is fine. Table 1 reports a tannin content in the hexane and dichloromethane fractions, which does not seem correct to me. The method used to measure tannins tends to give false positives. But there is also the possibility that the plant may contain methoxylated flavonoids, which do appear in the aforementioned fractions. The authors should check this information against a chromatographic run of these fractions
For the results on the measurements of tannins, we can find tannins with nonpolar solvents, in particular when hexane is the first solvent used. However, the quantity remains low. It is true that it is surprising that we do not see tannins in polar solvents, but this may be due to the fact that this plant is poor in tannins. Moreover, this absence in the polar fractions is confirmed by UPLC analyzes. Since the non-polar fractions did not show an antioxidant activity, they were not analyzed by UPLC, so the question of the nature of these compounds (tannins, methoxylated flavonoids or others) remains unanswered and that may be the subject of another study.
A paragraph has been added in the discussion part:
Lines 196 : Unexpectedly, the total tannin assay led to detect the presence of a small fraction of tannins in the apolar fractions only. Since hexane and dichloromethane extraction were previously shown to extract tannins with a low efficiency [30, 31] the fact that no trace of tannin was detected in the polar fraction is puzzling. UPLC analysis confirmed the absence of tannins in the polar extractions, however, since UPLC analysis were not conducted on the apolar fractions, we cannot strictly confirm the nature of the compound which led to a positive result with the total tannin assay

Round 2
Reviewer 1 Report
I read with interest the revised version of the paper and I find that authors addressed many major concerns.
I think the paper is acceptable for the publication after some further minor changes.
- Use the same accuracy when results are reported.
For example table 1:
M 8.21
W 8.5 (correct in 8.50)
then:
0.5 +/- 0.03 (correct in 0.50 +/- 0.03)
- Check and change text editing and spelling errors
- Captions should be improved better describing main features reported and thus leading the reader to have a self-explanatory graphical summary of results.
- In discussion I truly suggest to improve the paper trying to link phytochemical and antioxidants results here obtained with traditional and potential future use of the species.
Author Response
J'ai lu avec intérêt la version révisée de l'article et je trouve que les auteurs ont abordé de nombreuses préoccupations majeures.
Je pense que l'article est acceptable pour la publication après quelques modifications mineures supplémentaires.
- Utilisez la même précision lorsque les résultats sont rapportés.
Nous avons suivi cette recommandation et corrigée dans le texte (police verte et grasse)
- Check and change text editing and spelling errors
Another round of text editing has been realized which allowed to find some additional errors, now fixed (green and bold police).
- Captions should be improved better describing main features reported and thus leading the reader to have a self-explanatory graphical summary of results.
The caption of figure 1 has been added with explanation.
- In discussion I truly suggest to improve the paper trying to link phytochemical and antioxidants results here obtained with traditional and potential future use of the species.
On a le sentiment que cet aspect est déjà présent dans la version précédente (paragraphes des lignes 214 à 283). Aller plus loin dans la discussion peut apparaître comme spéculatif. Nous avons cependant modifié certaines phrases afin de mettre en évidence (i) la possibilité d'utiliser Phagnalon saxatile comme source d'acides chlorogéniques et (ii) le potentiel de cette plante à développer des aliments fonctionnels. A cet égard, les dernières phrases des concluons ont été clarifiées (police verte et audacieuse).

Reviewer 2 Report
the authors have welcomed the observation, answered my doubts, therefore I have not other comments, and I am satisfied.
Author Response
thank you for your satisfaction